# Exploring the Aesthetic and Functional Aspects of Recycled Furniture in Promoting Sustainable Development: An Applied Approach for Interior Design Students

Khaled Al-Saud [1], Rommel AlAli [2,*] , Adab M. Al saud [3], Amira S. Abouelela [4] , Rami Taha Shehab [5], Dalia Ali Abdel Moneim [4,6] and Alaa Eldin. M. Hamid [1]

1 Department of Art Education, King Faisal University, Al Hofuf 31982, Saudi Arabia; kmsoud@kfu.edu.sa (K.A.-S.); ahamid@kfu.edu.sa (A.E.M.H.)

2 The National Research Center for Giftedness and Creativity, King Faisal University, Al Hofuf 31982, Saudi Arabia

3 Educational Administration, World Islamic Sciences and Education University, Amman 11947, Jordan; adab.alsoud@wise.edu.jo

4 Art Education Department, College of Education, King Faisal University, Al Hofuf 31982, Saudi Arabia; aabouelela@kfu.edu.sa (A.S.A.); dabdelaziz@kfu.edu.sa (D.A.A.M.)

5 Information System Department, College of Computer Sciences & Information Technology, Al-Ahsa 31982, Saudi Arabia; rtshehab@kfu.edu.sa

6 Department of Industrial Design, The High Institute of Applied Art, 6 October, Ministry of Higher Education, Cairo P.O. Box 11728, Egypt

* Correspondence: ralali@kfu.edu.sa

**Abstract:** This study aimed to explore the sustainable aesthetic and functional dimensions of environmental waste in the context of interior design applications by students. Employing both descriptive and applied methodologies, a series of artistic works derived from environmental waste, specifically metal and wood, were conceptualized and executed by art education students. These works, totaling 11 artistic models in interior design, underwent chemical treatment as part of the process. An evaluation card, assessed by arbitrators, was utilized to gauge the standards of aesthetic and functional sustainability inherent in the artworks. Technical data were collected and subsequently analyzed using SPSS software, which facilitated the calculation of arithmetic averages, standard deviations, and *t*-tests to ascertain the extent to which sustainability standards were met within the aesthetic and functional dimensions of the works. The study findings indicated that the average response scores for the aesthetic and functional dimensions, pertaining to the achievement of sustainability for wood and metal waste, were notably high. This underscores the potential of producing artful works suitable for interior design applications within the spaces of the College of Education. With an average score of 3.984, students exhibited positive engagement with the aesthetic and functional aspects of their artistic products, indicative of their considerable ethical significance. This augurs well for the feasibility of attaining sustainability through the recycling of wood and metal waste. Furthermore, this research underscores the necessity of integrating aesthetic, environmental, and social values in achieving sustainable aesthetic and functional environmental values within the interior design curriculum. This integration demands a comprehensive understanding of user expectations, technological advancements, and the cultural background, customs, and traditions of both users and society at large.

**Keywords:** wood waste; metal waste; sustainability; interior design

## 1. Introduction

The rapid economic and population growth witnessed in recent decades has led to the depletion of numerous natural resources, resulting in significant environmental and societal consequences. This has prompted many countries worldwide to prioritize sustainability

to mitigate the negative impacts on the environment and human well-being. However, in developing countries, there exists a notable disparity between developmental practices and sustainable principles, underscoring the need for greater attention and clarity in defining sustainability [1]. Addressing these challenges requires the proactive identification of issues and the development of appropriate solutions before they escalate in complexity and cost.

Environmental concerns have increasingly garnered global attention, prompting countries to adopt environmental policies aimed at achieving sustainable development. This involves striking a balance between the urgent demands of economic and social progress and the imperative to preserve natural resources for future generations. Recycling environmental waste plays a crucial role in sustainability efforts, encompassing environmental, economic, and social dimensions. The continuity of natural resources is paramount for environmental preservation and societal well-being, necessitating multifaceted approaches to addressing environmental challenges. Environmental art has emerged as a promising avenue for sustainable development, offering diverse opportunities and options when integrated into sustainability initiatives [2].

This study investigates the use of environmental art as a strategy for preserving and fostering sustainable development. It assesses the significance of student projects and artistic endeavors in the field of interior design within the College of Education at King Faisal University. By utilizing recycled materials derived from environmental waste, both functionally and aesthetically, this study aims to promote sustainability by conserving natural resources and mitigating environmental impacts. Additionally, the design and production of artistic works using recycled environmental waste serve to educate the community about sustainability through creative expression. The practical aspect of designing a collection of artworks is expected to enhance the aesthetic appeal of internal and external environments, potentially contributing to a positive atmosphere for individuals and spaces.

Furthermore, this study explores the economic benefits of investing in waste recycling and environmental sustainability, highlighting the potential for positive returns while engaging students in creative design practices and leveraging environmental raw materials. It seeks to promote the adoption of best environmental practices in the field and generate actionable insights to inform future strategies. Data collection involves gathering input from students involved in artistic projects, as well as experts and interior designers tasked with evaluating these initiatives. Through this comprehensive approach, this study endeavors to advance sustainable development goals and foster a culture of environmental stewardship within the educational setting.

### 1.1. Research Problem

Applied arts serve multiple purposes, aiming to ignite the imagination of aspiring designers while fostering their personal skills and unique tastes. They play a pivotal role in enhancing both the aesthetic appeal and functionality of spaces, thereby instilling a sense of positivity in individuals. In recent years, there has been growing emphasis on environmental sustainability, prompting a shift towards repurposing waste materials through recycling initiatives. This process, which encompasses economic, aesthetic, and functional aspects, not only minimizes the depletion of raw materials but also contributes to environmental cleanliness. Within this context, interior design emerges as a key player in promoting environmental consciousness among the populace, including students. It encourages innovative approaches to utilizing and repurposing redundant materials into artistic creations, thereby addressing environmental challenges while simultaneously deriving value from waste. The rapid advancement of science and technology, coupled with the escalating population density, has resulted in a surge in consumable materials and solid waste across various forms and types.

Recognizing the need to address this issue, researchers have identified an opportunity to repurpose and rehabilitate these waste materials, especially considering their accessibility to students and their availability in diverse forms within the educational environment.

As interior design students routinely engage with a multitude of materials and resources in their projects, they are naturally inclined to explore ways to minimize costs, reduce financial burdens, and uphold environmental preservation while producing aesthetically pleasing and functional designs.

Hence, our research problem stems from the imperative to achieve environmentally sustainable interior design solutions for designated spaces within educational institutions. This goal can be accomplished by integrating the dimensions and principles of sustainable environmental design into the conceptualization and execution of interior space treatments and furniture designs. Consequently, the central question driving this study is as follows: To what extent do the works of art education students in interior design rely on end-of-use materials and repurposing them into artistic creations that embody the aesthetic and functional essence of educational spaces?

### 1.2. Research Questions

1.  What are the viable wood and metal environmental waste materials that can be repurposed to create artistic works in the interior design course?
2.  To what extent do the artistic works produced by interior design students meet the established sustainable aesthetic and functional standards for environmental waste utilization?
3.  Are there statistically significant differences, at a significance level of $\alpha \geq 0.05$, between the aesthetic and functional dimensions of furniture elements implemented in the interior design course by art education students?

### 1.3. Study Hypotheses

1.  It is hypothesized that the set of sustainable aesthetic and functional standards for environmental waste is not fully achieved in the applications of interior design course students.
2.  It is hypothesized that there is statistical significance regarding the sustainability criteria of the aesthetic and functional dimensions of the furniture elements implemented in the interior design course by art education students at the level $\alpha \geq 0.05$.

### 1.4. Study Objectives

1.  To inventory waste environmental materials suitable for producing furniture items in the interior design course.
2.  To utilize waste environmental materials, specifically wood and metal, in designing and implementing a group of furniture elements within the interior design course.
3.  To assess the level of sustainability standards for both the aesthetic and functional dimensions of waste environmental materials used in designing a set of furniture elements for interior design.
4.  To advance the understanding of the principles and dimensions of sustainable environmental design and apply them to the development of sustainable interior design solutions for various spaces and environments.

### 1.5. Importance of This Study

The significance of this study arises from its exploration of sustainable aesthetic and functional environmental design principles within interior design, which offers avenues for enhancing the internal environment of such spaces. By identifying alternative solutions and addressing design challenges, this study aims to improve the quality of educational services and enhance both aesthetic and functional outcomes, alongside promoting environmental, economic, and social sustainability. Furthermore, the importance of this research lies in understanding the sustainable values inherent in recycling environmental waste in interior design applications, carrying several theoretical and practical implications:

1.  Addressing a pressing issue: The utilization of environmentally friendly materials like wood and metal in recycling efforts can significantly mitigate visual pollution and en-

hance environmental sustainability. By spotlighting this issue, this study underscores the importance of developing effective interventions to address environmental waste from both aesthetic and functional perspectives.

2. Exploring a new approach: By focusing on experimentation with environmental waste, this study introduces a novel treatment method. By assessing the efficacy of this approach, this research expands the repertoire of strategies available for tackling environmental waste challenges within interior design.

3. Enhancing creativity and self-expression: Recycling environmental waste provides a platform for designers to express creativity and innovation. By leveraging environmental waste as an aesthetic medium, this study not only aims to improve aesthetic outcomes but also to enhance individual well-being and optimize the utilization of waste materials.

4. Potential cross-cutting benefits: Recycling environmental waste can yield broader positive effects on environmental sustainability. By producing furniture items through art and interior design, this study may enhance the production of recycled furniture, foster problem-solving skills related to environmental issues, and promote social interactions, beyond the immediate goal of waste treatment.

5. Practical implications for environmental intervention and remediation: This study's findings may have practical implications for interior designers, as well as professionals in economic marketing and environmental conservation. If this research demonstrates the efficacy of recycling environmental materials based on stakeholder feedback, it could inform the development of evidence-based methods and programs, guiding practitioners in implementing effective strategies to address environmental challenges and visual pollution.

6. Fostering responsibility and engagement in interior design for environmental sustainability: We aim to enhance cultural awareness regarding environmental sustainability within the field of interior design, while inspiring students and design enthusiasts to generate innovative ideas that integrate recycling in a manner that is both functional and aesthetically suitable.

7. Non-project initiative—empowering solutions for social and environmental impact: Our organization is committed to endorsing initiatives and projects that prioritize societal welfare and environmental conservation, with a strong focus on attaining long-term environmental sustainability.

Overall, the importance of this study lies in its potential to contribute to the understanding of addressing and recycling environmental waste, exploring innovative intervention approaches for environmental conservation and sustainability, fostering creativity and self-expression among interior designers, and providing practical insights for professionals working at the intersection of art and environmental conservation.

*1.6. Study Limitations*

1. Objective Limits: This study is confined to experimenting with waste environmental materials, specifically wood and metal, and implementing artistic works within the interior design course curriculum.

2. Time Constraints: This research is restricted to the duration of the second semester of the academic year 2023/2024, imposing limitations on the timeframe available for data collection and analysis.

3. Spatial Boundaries: This study is geographically constrained to the environment of the Al-Ahsa Governorate, specifically within the College of Education at King Faisal University. This limitation confines the scope of this research to a specific geographical area, potentially limiting the generalizability of the findings.

4. Human Limitations: This study's human subjects are restricted to students enrolled in the art education program at King Faisal University who are participating in the interior design course, as well as arbitrators responsible for evaluating artistic

products. This limitation restricts the sample population to a specific demographic, potentially impacting the diversity of perspectives represented in this study.

### 1.7. Study Terminology

Aesthetic and Functional Dimensions: As highlighted by Al-Jawfi, the interplay between function and beauty constitutes a fundamental objective of artistic endeavors. The artistic value is intrinsically tied to the functionality of the work, while also preserving its aesthetic aspects [3]. In the context of this study, aesthetic and functional dimensions refer to a set of elements and principles within artistic works that contribute to both visual appeal and practical utility in interior design, particularly through the recycling of wood and metal environmental waste.

Environmental Waste: Defined by Al-Eid as natural and industrial materials of diverse shapes and compositions that are haphazardly deposited in nature and can be repurposed and utilized [4]. In this study, environmental waste specifically pertains to the residual waste of wood and metal raw materials, as depicted in Figures 1–4 within the Al-Ahsa environment, which will be creatively repurposed in interior design projects.

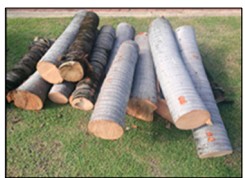

**Figure 1.** Wood.

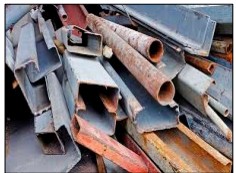

**Figure 2.** Metals.

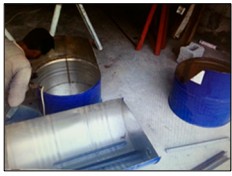

**Figure 3.** Metals.

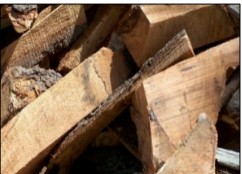

**Figure 4.** Wood.

Sustainability: Sustainability, as elucidated by Calcutawi [5], denotes the continuity and longevity of maintaining quality of life by harmonizing with the environment and utilizing natural resources for an extended duration. It encompasses vital processes that sustain life forms and their growth without depleting natural resources or causing harm to the environment. According to Al-Raddadi, sustainability entails the perpetuation of available materials and activities that meet the present generation's needs while enhancing

living standards without adverse environmental impacts [6]. In the procedural context of this study, sustainability refers to the enduring stability and longevity of the aesthetic and functional dimensions in interior design products crafted from wood and metal raw material residues, as depicted in Figures 1–4.

*1.8. Literature Review*

Several studies have emphasized the significance of recycling raw materials and promoting environmental sustainability to meet modern development needs while contributing to environmental cleanliness and economic viability. Al-Saud et al., conducted a study focusing on the utilization of lean waste through recycling as art pieces and decorative elements to enhance sustainability and environmental preservation in tourist destinations. Employing both descriptive and applied approaches, the study involved creating artworks from palm fronds, chemically treated by art education students, resulting in eight pieces. A questionnaire was utilized to assess the sustainability standards achieved by these artworks, with feedback gathered from 55 visitors to tourist sites attending an art exhibition at King Faisal University's Department of Art Education. Data analysis, employing SPSS (v25), included calculating averages and standard deviations and conducting *t*-tests to gauge the artworks' adherence to sustainability criteria. The findings indicated a high degree of sustainability achieved with palm waste, with positive visitor responses, suggesting potential for its use in complementing interior designs and decorations at tourist resorts. The study recommended further research to explore palm waste utilization in design and decoration beyond tourist sites [7].

Sarah et al., investigated the significance of interior designers in preserving nature reserves and expressing their identity through interior design and furniture. The research employed a combination of theoretical, descriptive analytical, and experimental methodologies to examine the applicability of proposed design steps. Design concepts for interior spaces and select furniture pieces were implemented and tested within the reserve to assess their effectiveness. The findings demonstrated the pivotal and influential role of interior designers in conservation efforts within the reserve through the implementation of sustainable designs. Furthermore, the study highlighted the ability of interior designers to showcase the identity and distinct characteristics of the reserve, whether environmental or otherwise. The utilization of recycled materials, such as palm waste, emerged as a desirable and natural sustainable option suitable for various applications within the Nabq Reserve [8].

Ahmed explored the potential of utilizing plastic waste in sculpture works through melting and assembly methods, assessing its contribution to enhancing artistic form and content. Employing descriptive, analytical, and practical approaches, the study demonstrated the feasibility of incorporating plastic waste into sculpture works, highlighting how plastic properties imbue unique sensory values and foster innovation and creativity [9]. Shafi'i and Al-Harbi investigated the sustainability and environmental preservation aspects of recycling natural resource waste in clothing products, particularly palm waste repurposed into fashionable belts. Employing descriptive and applied methodologies, the study designed and evaluated a series of belts, demonstrating high sustainability achievement scores, indicative of the viability of palm waste recycling in fashion accessory production [10].

Bahloul and Sarah underscored the imperative of integrating sustainable development dimensions into solid waste recycling practices to mitigate environmental and health hazards. The study emphasized the significance of scientific waste management methods in fostering local economies and achieving social, environmental, and economic balance [11]. Al-Jawfi explored the aesthetic and functional dimensions of environmental waste in artworks produced by students of the Art Education Department. Utilizing a descriptive and analytical approach, the study highlighted students' innovative use of materials to create aesthetically pleasing artworks, emphasizing the educational and environmental significance of reusing materials [3].

Naseer proposed a future vision for recycling agricultural waste to furnish interior spaces and address interior design challenges. Through a descriptive analytical approach, the study advocated for utilizing agricultural waste recycling techniques to conserve energy, create environmentally compatible interior spaces, and mitigate pollution rates, thereby enhancing public health and resource sustainability [12]. Al-Kandari investigated contemporary methods for utilizing waste recycling to create innovative artistic sculptures, aiming to enhance students' creative abilities in producing artistic models. Employing an applied descriptive method, the study demonstrated the potential for improving students' innovative skills by harnessing waste materials to generate innovative artistic models [13].

In a study conducted by Al-Khalidi, the focus was on the role of field training for interior design students in achieving comprehensive development in the United Arab Emirates. The research highlighted the dynamic nature of the arts, characterized by continuous internal and external renewal. The study findings revealed that despite the parallel relationship between political stability, economic prosperity, and development, the progress in the field of interior design is intricately tied to experiential learning through field training. The practical experiences gained during field training played a crucial role in enhancing competencies and skills, contributing to developmental advancements. Additionally, the study emphasized the significance of well-designed field training programs and their successful implementation within executive offices and institutions, highlighting their impact on the overall strength of academic programs in interior design [14].

In a study conducted by Al-Baghdadi, the focus was on maximizing the utilization of waste wood products to develop new plastic formulations for sustainable three-dimensional materials in photography. The research specifically examined the availability of waste wood raw materials in the Damietta Governorate. Employing a descriptive and experimental approach, the study included artistic experiments conducted with third-year students from the Department of Art Education at the Faculty of Specific Education in Damietta. The experiments involved creating artworks using wood waste with regular, irregular, and organic geometric forms, as well as producing a collective mural made from wood waste. The findings highlighted the potential of wood waste, not only in terms of its shape but also in creating diverse surface values, particularly when combined with colors and dyes. This enhanced the expressive aspect of both individual student works and collaborative wall paintings. The study emphasized the importance of utilizing wood waste, due to its multifaceted capabilities and formulations, in the field of photography [15].

Upon examining previous studies, several notable patterns emerge. Some of these studies have delved into the effectiveness of art-based programs in repurposing environmental materials, showcasing a growing interest in leveraging creative strategies for sustainability. A common thread among these studies is their shared objective of finding innovative ways to recycle raw materials through experimentation, underscoring a collective effort to address environmental challenges. However, there is variability in sample size across these studies, with some focusing on student samples while others utilize self-experience or critical analysis methods. Methodological approaches also vary, with most studies employing quasi-experimental or descriptive methods to assess the effectiveness of art-based programs. In contrast, the current study adopts a comprehensive approach encompassing descriptive, analytical, and experimental methods. While previous research has addressed a range of environmental wastes, the current study specifically targets wood and metal materials and their constituents for a more focused investigation. Despite methodological differences, the current study shares a common goal with previous research: recycling raw materials. Drawing on insights from prior studies, the current research aims to refine sampling techniques, methodologies, and tool development to align with its specific objectives and population.

### 1.9. Theoretical Framework

First: Sustainability and environmental waste

Theoretical discussions and the educational literature underscore the critical importance of addressing sustainability concerns and managing environmental waste to safeguard public health and mitigate environmental risks. Scholars like Shams highlight the potential health hazards posed by unmanaged waste accumulation, emphasizing the need for proper waste disposal to prevent disease outbreaks [16]. However, defining waste proves complex, as what may be considered waste by some is deemed valuable by others, as indicated in studies by [7,17,18]. Consequently, efforts in waste management aim to mitigate its adverse effects on both the environment and society. Sustainable interior design emerges as a contemporary design philosophy that integrates environmental waste into design practices, aligning with Hussein's assertion that interior spaces should sustainably support human life, comfort, and stability while minimizing resource consumption [19]. The environment encompasses the natural and social systems surrounding individuals, impacting and being influenced by human activities, as discussed by [1,20]. This includes natural resources shaped by human interaction to fulfill societal needs, reflecting cultural and social dynamics, as emphasized by [21]. Understanding these concepts is integral to informing design practices that harmonize with both natural and human environments, facilitating sustainable development and cultural expression within communities.

Throughout history, human endeavors have centered on harnessing environmental resources to meet various needs, leading to a symbiotic relationship between humanity and its surroundings. The RSMA Design website highlights how human intervention drives advancements and environmental transformations, facilitating civilization's progress by augmenting control over natural elements [22]. Kirwan underscores that this dynamic equilibrium between humans and their environment fosters adaptability and innovation, enabling sustainable development. Sustainability is an overarching global responsibility that necessitates communication, continuity, and proactive responses to present circumstances, facts, and requirements, as well as future developments, while actively contributing to their advancement. This includes the development and enhancement of tourist destinations and environments, recognizing the associated needs and intertwining this progress with artistic creations that augment the aesthetic aspects of these surroundings [23]. Bakri highlights the definition provided by the United Nations Council on Sustainable Development (PCSD), which characterizes sustainability as an ongoing endeavor that seeks to improve the environment, society, and economy for the benefit of current and future generations, ensuring its transmission from one generation to the next [24].

Second: Recycling and its relationship with sustainability

The concept of recycling gained prominence during the 1930s and 1940s in response to the economic downturn, with materials like nylon, rubber, and metals being recycled, as noted by Helen et al. [25]. However, its popularity waned in the United States until the early 1970s, when it was reintroduced on Earth Day in 1970 and steadily gained acceptance over time, as highlighted by Ahmed [9]. Recycling involves collecting used materials, converting them into raw materials, and then reproducing them for reuse, according to Marwan, who notes its broad scope encompassing any reusable item [26]. This process effectively repurposes waste materials like iron, plastic, glass, and paper, thereby mitigating resource depletion from nature, as emphasized by Sanjak [27].

Numerous studies and initiatives, including those by [3,9,11,28–30], underscore the importance of recycling environmental waste and advocate for the establishment of specialized recycling facilities. This trend is further reinforced by the existence of four national companies in Al-Ahsa Oasis dedicated to studying and implementing large-scale investment projects for waste recycling. Notably, the National Waste Management Center, established in 2019, prioritizes waste management organization and investment stimulation through the circular economy framework [31]. National Waste Management emphasizes innovation adoption and modern technology utilization to achieve waste disposal via

recycling, alongside raising public awareness to reduce waste generation and promote reuse. Additionally, the center focuses on encouraging diverse investment models to ensure its financial sustainability [31].

The recycling process, as outlined by the US Environmental Protection Agency and detailed by Al-Huwaimani [32], comprises three key stages: collection and treatment, manufacturing, and consumer utilization. Recyclable materials undergo sorting, cleaning, and processing at specialized facilities to produce raw materials for manufacturing various products like newspapers, soft drink bottles, and carpets. Consumers then close the recycling loop by purchasing products made from recycled materials, thereby reducing costs and aiding in environmental preservation by curbing resource depletion.

Research by [9,16,29] highlights the environmental benefits stemming from recycling, particularly of metals and wood, including reduced carbon dioxide emissions during handling and burning processes, consequently mitigating global warming. Conversely, failure to recycle leads to increased air pollution from waste combustion, with paper and plastic burning being notable contributors. Moreover, improper waste disposal results in soil and water contamination due to toxic substance leakage. Ultimately, recycling plays a pivotal role in preserving natural resources and achieving environmental sustainability.

Third: Interior design and sustainable environment

The pursuit of sustainable environmental formation entails the multifaceted integration of diverse design systems with modern technological advancements. Its objective is to preserve natural energy reservoirs, ensure user comfort, mitigate pollution, and foster eco-friendly alternatives through recycling initiatives. Al-Houti underscores the foundational premise of environmentally sustainable design, emphasizing its intrinsic alignment with sustainability principles and its imperative utilization of environmental resources to safeguard future generations [33]. Pazzaglini highlights a prevailing shift towards incorporating sustainable environmental design attributes within interior design treatments, with a growing emphasis on materials capable of harmonizing with the environment to enhance user comfort across social, economic, and aesthetic dimensions [34].

Various conceptual frameworks have emerged to address the dimensions of sustainable development, encompassing economic, environmental, and social facets. AlAli and Aboud posits sustainability as a fundamental driver for ensuring enduring livelihoods, mitigating resource depletion, and fostering social stability [35]. Al-Sayed extends this discourse by acknowledging the myriad concepts and dimensions associated with sustainable development, including those pertinent to aesthetic functionality [36]. Interior design, as articulated by [23], has become a focal point for creative endeavors aimed at repurposing environmental waste into functional and aesthetic elements, thereby contributing to energy conservation, pollution reduction, and the production of eco-conscious artworks.

Considered an applied plastic art, interior design serves as a platform for nurturing creativity, fostering personal expression, and instilling an appreciation for three-dimensional compositions. Drawing insights from the works of [9,18,37,38], interior design emerges as a conduit for realizing sustainable environmental visions, aligning with contemporary trends, and addressing societal needs. This necessitates an adept understanding of materials, tools, and methodologies to harness the aesthetic and functional potential of environmentally sourced materials, thereby mitigating visual pollution while enriching spatial experiences.

Within educational settings, such as colleges, students are immersed in environments ripe for exploration and experimentation with environmental waste as a medium for artistic expression. Encouraged by the works of their predecessors, students are galvanized to repurpose environmental waste into three-dimensional artworks that enrich educational and recreational spaces within their institutions.

Furthermore, interior design serves as a catalyst for societal introspection and behavioral correction, inspiring individuals to embrace sustainable lifestyles and environmental stewardship. Visual artists, in their capacity as interior designers, wield their creativity to advocate for sustainability through imaginative interventions and innovative design approaches. By reimagining artistic mediums to embrace sustainability principles, they

contribute to a paradigm shift towards a more environmentally conscious and aesthetically pleasing existence.

In essence, interior design transcends mere spatial arrangement, evolving into a transformative force that not only enriches human experiences but also fosters environmental consciousness and societal responsibility. Through creative ingenuity and conscientious design practices, interior designers wield their craft to shape a more sustainable and aesthetically captivating world.

Fourth: The functional and aesthetic dimension in interior design

The primary objective of designed models is to fulfill their intended purpose effectively. Each product is imbued with a distinct function, encompassing not only its per-formative role but also various auxiliary functions, which may include symbolic, aesthetic, or sensory dimensions. Al-Saud et al., underscore the inherent complexity of human needs, wherein both aesthetic and functional considerations play pivotal roles. The relative im-portance of these aspects varies depending on the nature and intended usage of the product, underscoring the intrinsic link between its purpose and perceived value [7]. As elucidated by [29], the purpose or objective of a product holds significant importance, serving as a foundational goal that guides its design and development process.

Furthermore, the aesthetic dimension of a product engenders a profound sensory experience, eliciting feelings of admiration and contemplation. Beauty, in this context, is characterized by the harmonious interplay of form, texture, and visual appeal, evoking sensations of pleasure, enjoyment, and even emotional resonance. Objects and creations are deemed beautiful when they possess the capacity to captivate the soul, imbuing the observer with a profound sense of aesthetic gratification and resonance. This intertwining of form and function underscores the intrinsic allure of well-designed products, wherein aesthetic appeal intertwines seamlessly with utilitarian efficacy to evoke profound sensory responses.

Achieving the perfect equilibrium between aesthetic allure and functional efficacy within interior design parallels the intricate strokes of an artistic masterpiece. This synthesis of creativity and practicality crafts a comprehensive design that excels in both form and function. RSMA Design emphasizes the pivotal importance of discerning the dual objectives of aesthetics and functionality inherent to the space under transformation [22]. This necessitates a meticulous exploration of the envisioned purpose, meticulous attention to detail, and a thorough analysis of spatial dynamics, encompassing dimensions, overlaps, functional requisites, lighting schemes, ventilation systems, and other pivotal elements essential for optimal design.

In this pursuit, several key dimensions emerge as paramount:

1. Color Selection: Striking a delicate harmony between captivating hues and functional imperatives stands as a cornerstone. Colors must not only be visually appealing but also complement the intended practicality of the space. Additionally, materials chosen for their amalgamation of beauty, resilience, and ease of maintenance lay the foundation for exemplary interior design.

2. Furniture and Accoutrements: Furniture pieces serve as more than mere embellishments; they embody a fusion of aesthetic splendor and utilitarian comfort. Ensuring coherence with the overarching design style, furniture and accessories should enhance spatial comfort and flexibility and meet exacting standards of quality, durability, and environmental sustainability.

3. Strategic Lighting Implementation: Lighting intricately weaves strengths and weaknesses within interior design, casting an ethereal glow upon furnishings and paints from an aesthetic standpoint while providing functional illumination. A judicious selection of lighting sources ensures uniformity, balance, and visual comfort, while embracing energy-efficient practices and environmentally friendly energy sources.

4. Attention to Detail: In the minutiae of each element lies the hallmark of creativity and precision. From fabrics to fixtures, doors to windows, every detail demands

meticulous consideration to imbue harmonious enhancements that elevate both form and function while remaining true to the overarching design ethos.

5.   Iterative Design Refinement: Continual refinement is indispensable to the evolution of interior design, necessitating the exploration of diverse options tailored to the unique exigencies of the space and its stylistic inclinations. Through iterative experimentation and adaptation, a harmonious synthesis of aesthetic allure and functional efficacy is ultimately achieved.

In essence, achieving the ideal balance between aesthetic grandeur and functional pragmatism within interior design requires a nuanced understanding of spatial dynamics, meticulous attention to detail, and an unwavering commitment to iterative refinement. It is through this holistic approach that interior spaces transcend mere functionality to become enduring works of art, seamlessly blending form and function into a cohesive whole.

Fifth: Chemical treatments of wood and metal waste

The utilization of chemical treatments for wood and metal waste represents a critical avenue for enhancing their properties and extending their utility. With regards to wood, the addition of various chemicals serves to deter or prevent attacks from organisms that degrade wood fibers. Swedish Wood outlines that these chemicals can be applied manu-ally through painting or immersion, or industrially via pressure soaking [39]. However, it's noted that surface treatments such as coating and immersion have limited penetration into the wood, primarily safeguarding exposed surfaces. As exemplified by Bryan [40], the historical use of chromium arsenate in wood preservatives since the 1940s has been in-strumental in protecting against rot and pest infestations.

Similarly, mineral processing entails various chemical treatments aimed at extracting metals from ores. Euro training delineates two primary processes: physical concentration of ores and chemical metal extraction. These processes are often intertwined, with physical separation preceding chemical treatment in some instances. Chemical treatments for met-als encompass diverse techniques such as electropolishing, electroplating, etching, and galvanizing, all aimed at enhancing durability and corrosion resistance. Refs. [9,10,41] high-light the versatility of chemical treatments, which not only render waste materials safely disposable but also transform them into valuable products or artistic creations suitable for various environments.

Moreover, adherence to sustainable practices and local environmental regulations is imperative during waste treatment, as emphasized by [21]. Such measures ensure the con-tinuity and sustainability of treatment processes while enhancing aesthetic appeal and mitigating visual pollution in the surrounding environment. By incorporating chemical treatments into waste management strategies, industries can not only optimize resource utilization but also minimize environmental impact, fostering a more sustainable ap-proach to material processing and utilization.

Sixth: Technical uses of wood and metal waste

The utilization of recycled wood and metal materials within educational facilities holds promise as a means to augment interior design, potentially enriching students' quality of life while fostering environmental awareness. As underscored by Shafi'i and Al-Harbi, interior designers bear the responsibility of cultivating a sustainable environ-ment that prioritizes student comfort and well-being [10]. Integrating wood and metal waste into design endeavors presents a unique and innovative approach, imbuing spaces with natural beauty and artistic flair.

Embracing this creative concept may necessitate experimentation and exploration, but it can yield distinctive and captivating artistic creations. Studies such as Al-Baghdadi's and others illuminate artists' inclinations towards utilizing environmental materials in their natural or modified states, showcasing the transformative potential inherent in waste materials [15]. Artists like Pablo Picasso, John Rousseau Howard, and Louisa Nevel-son exemplify the transformative power of repurposing waste materials into high-level

artistic works, underscoring the need for a re-evaluation of arts education to incorporate sustainable practices and minimize environmental impact.

Numerous studies, including those by [2,10,42–44] Ibrahim and Zayed [42], Faraj [43], Shafi'i and Al-Harbi [10], and Ahmed and Aly [2], delineate various techniques for incorporating wood and metal waste into interior design, including the following:

1.  Plastic Arts Techniques: Utilizing wood and metal waste as three-dimensional elements can yield aesthetic furniture pieces such as chairs, cupboards, shelves, and multi-use tables.
2.  Ink or Color Drawing: Wood and metal textures serve as ideal bases for ink or color drawing, offering decorative potential as wallpapers or furniture.
3.  Sculpture: Carving or modeling certain wood and metal waste pieces can result in artistic sculptures that add a unique dimension to interior spaces.
4.  Mixing Media: Wood and metal waste can be combined with other artistic media, including plant-derived extracts, to enhance artistic effects and textures.
5.  Deletion and Addition Techniques: Incorporating wood and metal waste through deletion and addition techniques enables the creation of innovative visual effects, allowing waste materials to serve as integral components of artistic forms or backgrounds.

By harnessing the inherent aesthetic and structural qualities of wood and metal waste, interior designers can transcend conventional design paradigms, fostering sustainable and visually captivating environments conducive to holistic student experiences.

Seventh: Technical experiments with wood and metal waste

Theoretical frameworks and prior research underscore the manifold artistic possibilities inherent in repurposing wood and metal waste within interior design contexts. Utilizing recycled wood for furniture fabrication presents a sustainable alternative, imbuing spaces with a distinct aesthetic vocabulary while aligning with sustainable design principles. Moreover, this practice breathes new life into aged wood, mitigating the demand for virgin timber and thus contributing to forest conservation and wildlife habitat preservation.

Furthermore, the manufacturing process of recycled wood furniture entails lower energy consumption and reduced greenhouse gas emissions compared to crafting new furniture from raw materials. Notably, recycled wood furniture exhibits exceptional durability, standing the test of time with resilience against warping and cracking. Its versatility allows for seamless integration with various materials such as glass, metal, or leather, fostering an eclectic yet harmonious aesthetic.

The Idei.club website showcases compelling examples of artists' experimental endeavors in utilizing and repurposing wood and metal waste within interior design contexts, exemplified by Figures 5 and 6. These illustrations serve as tangible manifestations of the creative potential inherent in recycling waste materials, inspiring innovative design solutions that marry sustainability with artistic expression [45].

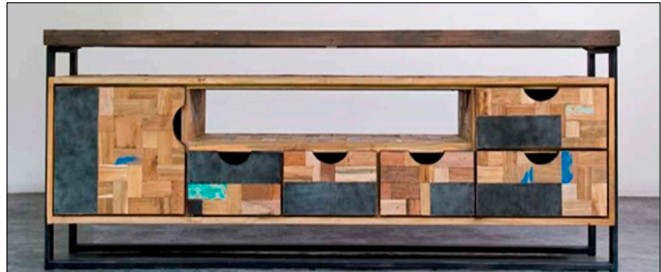

**Figure 5.** Furniture unit crafted from recycled wood waste.

Moreover, metal possesses inherent value and can undergo repeated recycling processes without compromising its quality. This recycling procedure involves the reuse, melting, and reshaping of scrap metal, thereby ensuring its continual reuse and return to the melting furnace. The recyclability of metal renders the process more convenient and

cost-effective than sourcing raw materials from the earth's crust. Utilizing recycled metals not only reduces production costs but also boasts environmental benefits.

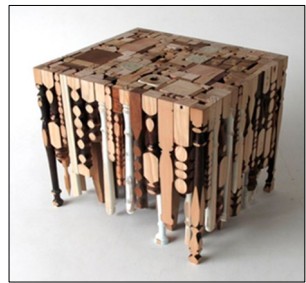

**Figure 6.** Recycled wood waste table.

Recycling metals minimizes the need for extracting raw materials from the earth, thereby mitigating the environmental impact associated with mining activities. This process avoids the disruption caused to natural landscapes during mining operations and helps preserve valuable resources in their natural state. Additionally, recycling metal consumes less energy compared to extracting and processing virgin materials, contributing to overall energy conservation efforts. Federalmetals presents examples of recycled metalwork, shown in Figures 7–9, showcasing the transformative potential of recycling in creating innovative and sustainable design solutions [46,47].

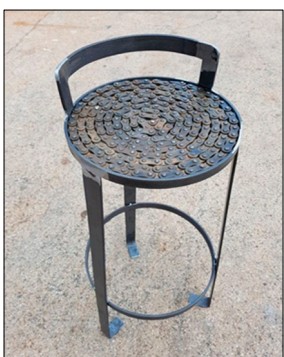

**Figure 7.** Chair crafted from recycled metal waste.

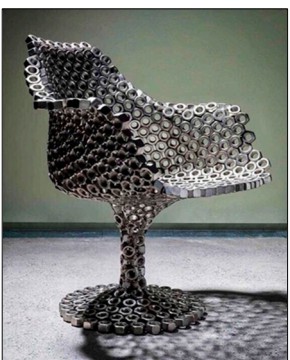

**Figure 8.** Chair fashioned from recycled metal waste.

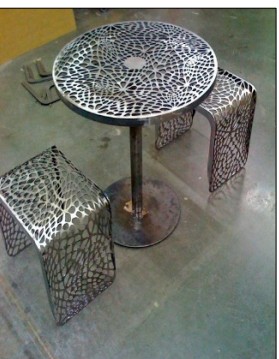

**Figure 9.** Table created from recycled metal waste.

## 2. Methods

### 2.1. Research Approach

This study employed a combination of experimental and descriptive analytical methods, tailored to the nature and objectives of the research. The experimental method facilitated the control of variables, with the exception of one variable, which the researchers manipulated to ascertain and evaluate its influence on the phenomenon under investigation. This approach allowed for a comprehensive examination of the research subject while providing valuable insights into its underlying dynamics.

### 2.2. Study Population and Sample

This study focused on a sample population comprising 30 students from the Department of Art Education at King Faisal University who were enrolled in the practical training for the interior design course. The sample was purposively selected to select a subset of students' works, resulting in a collection of 11 artistic designs created collaboratively. These designs utilized discarded wood and metal materials with the objective of recycling them and exploring their potential application in interior spaces as complementary elements to interior design. The selected sample was deemed representative of all students enrolled in the Department of Art Education at the university.

### 2.3. Study Tool

This research employed a comprehensive art analysis card comprising two primary dimensions: beauty and function. This card was designed to assess the utilization and sustainability of wood and metal waste recycled for interior design purposes and their integration into interior spaces. It comprised six criteria aimed at evaluating the efficacy of repurposing these materials. Table A1 provides a detailed depiction of this card alongside its criteria.

#### 2.3.1. Validation of the Sustainability Scale

To ensure the validity and reliability of the analysis card, the researchers subjected it to scrutiny by a panel of experts subsequent to formulating its axes and statements. The panel comprised 10 specialists in the fields of art and interior design. This rigorous validation process aimed to ascertain the experts' consensus regarding the suitability, comprehensiveness, and accuracy of the scale for evaluating the utilization of wood and metal waste in interior design. Specifically, the experts evaluated:

1. The applicability of the scale to the study sample.
2. The inclusivity of the scale in encompassing criteria and axes pertinent to working with wood and metal waste.
3. The precision of the scale's wording and its suitability for practical application.

Based on the experts' feedback, necessary modifications were made to the observations, resulting in a remarkable agreement rate of 95% among the experts. This high level of consensus further bolstered the validity and reliability of the sustainability scale.

### 2.3.2. Reliability of the Sustainability Scale

The reliability coefficient of the criteria within the art analysis card, pertaining to the aesthetic and functional sustainability dimensions of wood and metal utilization in interior design, was computed. This evaluation employed the test–retest method, involving a sample of 5 individuals distinct from the study participants. The test–retest approach involved administering the scale twice, with a two-week interval between administrations. The Pearson Correlation Coefficient was utilized to quantify the degree of consistency between responses from the two test instances. The resulting coefficient value was determined to be 97%, indicating a high level of reliability. This coefficient was deemed adequate for the purposes of this study, ensuring the reliability of the sustainability scale in evaluating wood and metal usage within interior design contexts.

### 2.3.3. Correcting the Sustainability Scale

The finalized version of the scale comprised 6 criteria, encapsulating the dimensions of beauty and function, as illustrated in Table A1. Three department members, specialized in scientific training and interior design, conducted the analysis of the interior design artworks. They assessed each artwork based on a five-point scale, where the judges selected one of five alternatives: excellent, very good, good, acceptable, or weak. Assigning grades from 1 to 5 corresponded to the statements in sequential order, resulting in a maximum score of 30 for each product and a minimum score of 6 for the judges' responses.

The total of 30 points was distributed across the two dimensions of the product analysis card, with each dimension comprising 15 points. The researchers categorized the judgments on the scale items into three levels: weak, medium, and high. These categorizations were determined based on the highest numerical value (5) and the lowest value (1) on the five-point scale. The difference between these values was divided by 3 to establish the range for each level. The weak level spanned from 1 to 2.33, the medium level ranged from 2.34 to 3.67, and the high level encompassed scores between 3.68 and 5.

### 2.4. Study Procedures

1. Field Visit: Conducted a field visit to sites where environmental waste is collected to identify waste wood and metal materials.
2. Practical Applications: Executed practical applications utilizing waste wood and metal materials in artistic works. Additional materials such as adhesives, staples, fabrics, and industrial paints were incorporated to enhance aesthetic aspects and ensure proper implementation.
3. Sustainability Standards: Established sustainability standards and devised an analysis card to assess the degree to which the implemented designs met these standards for wood and metal waste.
4. Validation of Scale: Validated the scale by obtaining feedback from arbitrators to ensure the appropriateness, clarity, and alignment of the statements with the study objectives and procedures.
5. Artwork Display: Showcased executed artworks in the College of Education lobby, allocating specific spaces for each work alongside complementary accessories like paintings and advertisements. Specialists from the Art Education Department then analyzed the artworks using the prepared analysis card.
6. Data Analysis: Utilized the SPSS program to transcribe and statistically analyze the data. Calculated frequencies, arithmetic means, and standard deviations to evaluate the criteria for achieving sustainability for wood and metal waste. Additionally, computed average scores for each dimension of beauty and function separately.
7. Presentation and Discussion: Presented and discussed the results, designing tables and incorporating images of furniture elements as exemplars of executed designs resulting from the recycling of wood and metal waste.

*2.5. Statistical Processing*

To address the research questions, various statistical methods were employed to gauge the perspectives of the study sample regarding the viability of utilizing wood and metal materials within interior design for interior spaces. Specifically, the following statistical techniques were utilized:

1.  *t*-test: This statistical test was utilized to compare the means of two groups and determine whether there was a significant difference between them in terms of their opinions on the usage of wood and metal materials in interior design.
2.  Arithmetic means: Arithmetic means were calculated to determine the average opinion of the study sample regarding the feasibility and effectiveness of incorporating wood and metal materials in interior design projects.
3.  Standard deviations: Standard deviations were computed to assess the dispersion or variability of opinions within the study sample regarding the utilization of wood and metal materials in interior design. This provides insights into the consistency or diversity of opinions among participants.

**3. Results and Discussion**

This research revealed that an effective strategy for attaining sustainable aesthetic and functional values in interior design involves strategic planning that harnesses elements and materials from the local environment. This approach guides the design process towards fulfilling sustainability requirements and fosters a continuous future-oriented vision aimed at achieving sustainable aesthetic and functional environmental values within the interior design curriculum at the Department of Art Education, College of Education. This approach influences design decisions and implementation methods from the outset.

Regarding the first research question—"What is the environmental waste of wood and metal that can be used to produce works of art in the interior design course?"—the researchers employed various sources and procedures, including the following:

Firstly, we established reference studies and a theoretical framework: Drawing upon the existing literature and theoretical frameworks, the researchers identified a range of environmental wastes suitable for recycling into new and alternative products. These wastes encompassed the following:

*   Wood waste, including tree waste, various industrial wastes, and construction and urban waste.
*   Mineral waste, such as wastes derived from petroleum or liquids.
*   Metal waste ranging from factory-related metal waste to post-consumer environmental metal waste, including cans and other everyday metal pieces.

These findings underscore the potential of utilizing diverse wood and metal waste materials to realize the concept of sustainability within interior design practices.

Secondly, field visits were conducted to various locations as part of the research methodology.

Thirdly, reference comparisons and additional field visits were utilized to identify specific environmental waste materials that align with the principle of sustainability and are suitable for creating artistic products in interior design. These waste materials specifically include wood waste from construction, trees, and industries, represented by shapes in Figures 10–12, as well as wood waste and metals from barrels, factories, and the environment, represented in Figures 13–15. The availability of these raw materials, namely wood and metal waste, can be attributed to their presence in the environment. These materials can be easily utilized in the production and implementation of diverse artistic works. These works can serve both aesthetic and utilitarian purposes, such as enhancing interior spaces with low-cost furniture items like seats, tables, swings, storage cupboards, lighting units, and more. Additionally, recycling these materials not only contributes to sustainability but also offers significant aesthetic and design value while maintaining durability. Moreover, other materials, such as colors, can be incorporated to further enhance their visual appeal.

The findings from this research align with previous studies that have explored the utilization and application of various design practices. Notably, studies conducted by Shafi'i and Al-Harbi [10], Sarah et al. [8], and Bahloul and Sarah [11] have similarly addressed the uses and employment of materials in different design fields.

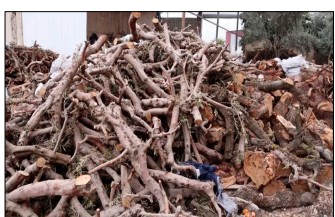

**Figure 10.** Wood waste from trees.

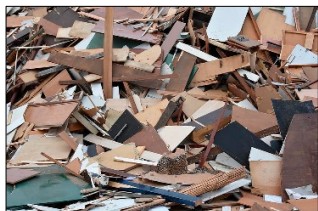

**Figure 11.** Industrial wood waste.

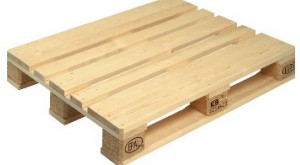

**Figure 12.** Wood waste from construction.

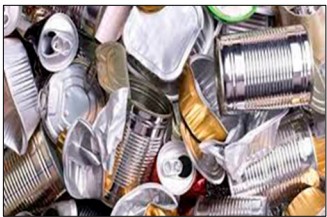

**Figure 13.** Environmental consumption.

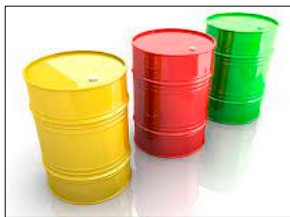

**Figure 14.** Metal waste in barrels.

Regarding the results pertaining to the second question, which focuses on the extent to which a set of sustainable aesthetic and functional standards for environmental waste is achieved in the applications of interior design course students, the researchers employed several methods for analysis.

To address this question, models of artistic products created by interior design course students were showcased in the college's lobbies. These models were then subjected to

analysis by experts in the field. Additionally, a statistical analysis was conducted using arithmetic averages to assess the degree to which these waste materials contribute to the implementation of furniture elements in interior design. Diagram is presented in Figure 16 and visual representations of the findings can be observed in Figures 17–29. Furthermore, Table 1 presents the arithmetic means and standard deviations of the aesthetic and functional dimensions of wood and metal waste in the products created by interior design course students.

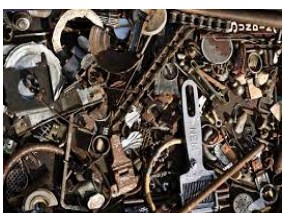

**Figure 15.** Industrial metal waste.

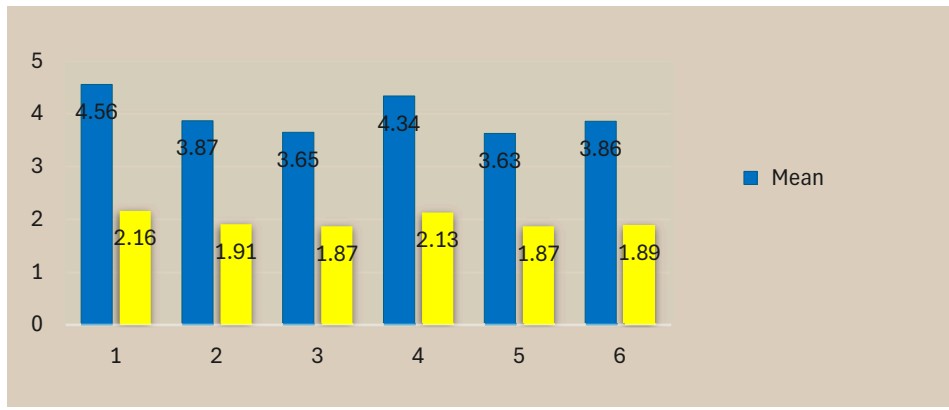

**Figure 16.** Mean and standard deviation of scores for aesthetic and functional dimensions in interior design standards.

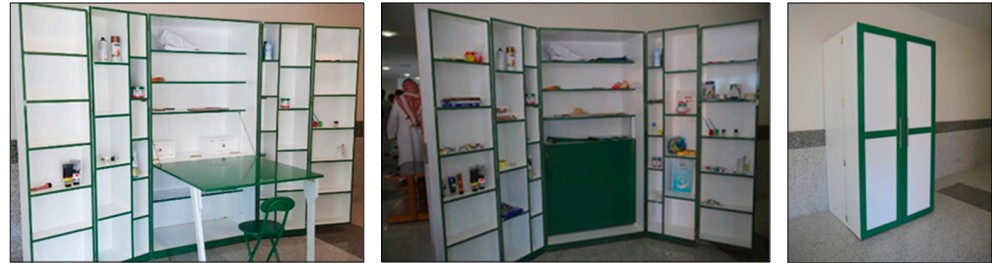

**Figure 17.** Utilization of wood waste in a storage cupboard for artistic tools for students of the Art Education Department.

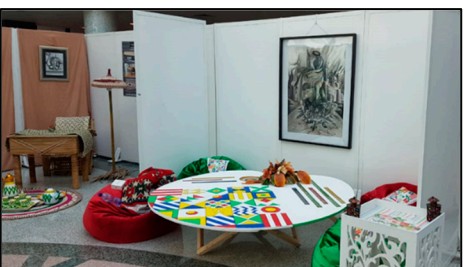

**Figure 18.** Low circular table constructed from recycled wood.

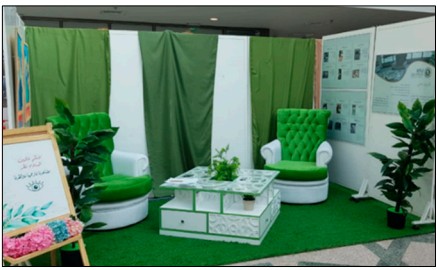

**Figure 19.** Seating units crafted from recycled wood.

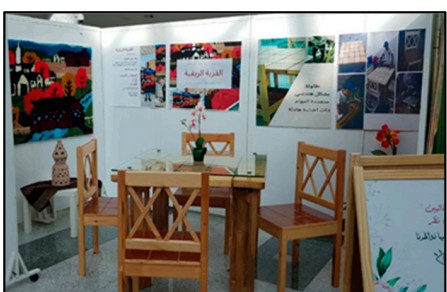

**Figure 20.** Elongated table with four wooden chairs.

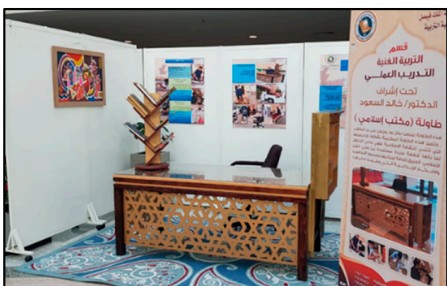

**Figure 21.** Wood-design office environment.

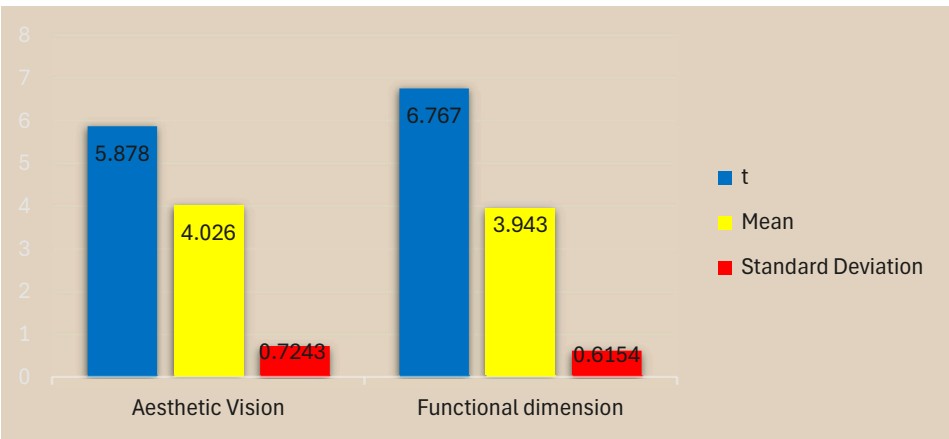

**Figure 22.** Mean, standard deviation, and t-value analysis of specialists' responses on technical product analysis cards in interior design.

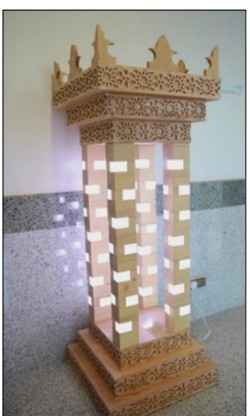

**Figure 23.** Wooden waste processed into lighting unit.

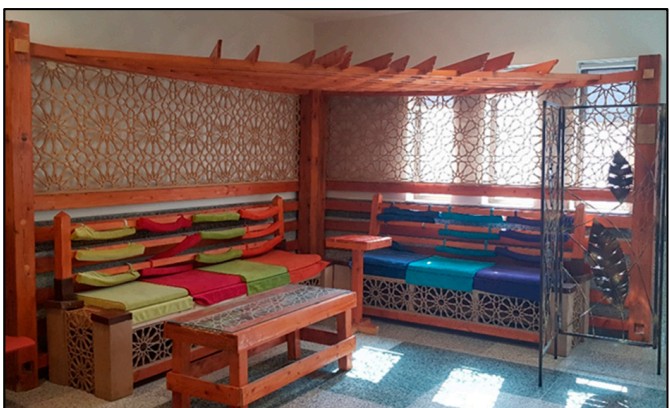

**Figure 24.** Recycled seating constructed from wood and fabric waste.

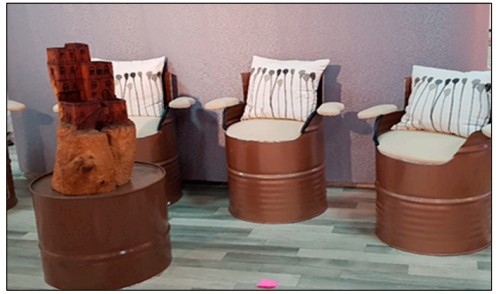

**Figure 25.** Seating units crafted from metal waste (barrels).

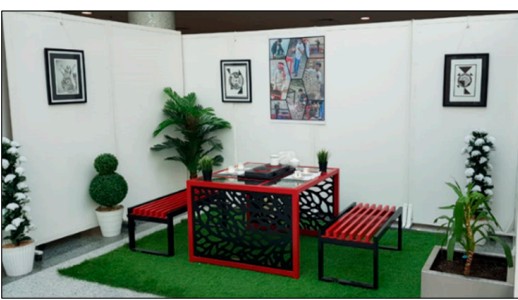

**Figure 26.** Table crafted from metal and wood.

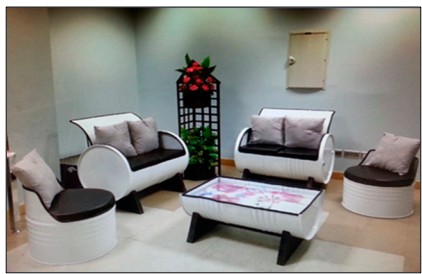

**Figure 27.** Seating units enhanced with leather and wood crafted from metal waste (barrels).

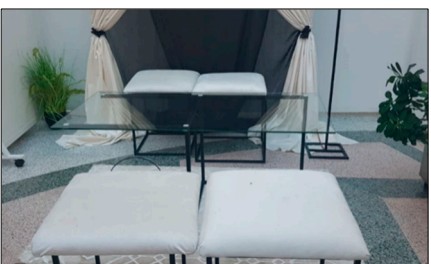

**Figure 28.** Seating units crafted from metal waste.

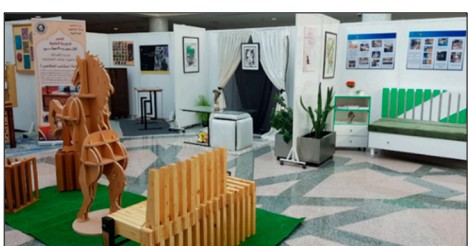

**Figure 29.** Visualization of works crafted from wood and metal waste.

**Table 1.** Mean and standard deviation analysis of aesthetic and functional dimensions for wood and metal waste in interior design students' projects.

| No. | Fields | Mean | Standard Deviation | Contribution Level |
|:---:|:---|:---:|:---:|:---:|
| 1 | Characteristics of the artwork material | 4.56 | 2.16 | High |
| 2 | Varied utilization of artistic elements and principles | 3.87 | 1.91 | High |
| 3 | Incorporation of multiple materials in the artwork | 3.65 | 1.87 | Medium |
| 4 | Proficiency in executing intricate technical details | 4.34 | 2.13 | High |
| 5 | Alignment of form and function | 3.63 | 1.87 | Medium |
| 6 | Precise execution of artwork details | 3.86 | 1.89 | High |
|  | Total | 3.984 | 1.972 | High |

Based on the findings presented in Table 1 and Figure 16, it is evident that wood and metal materials have made a significant contribution to the recycling and implementation of furniture elements in interior design, specifically within interior spaces. The overall score for the aesthetic and functional dimensions of wood and metal materials was notably high, reaching an arithmetic average of 3.984. Although the criteria of material diversity and the alignment of form with function achieved slightly lower average contributions of 3.65 and 3.63, respectively, these figures still indicate the successful utilization of wood and metal waste materials in interior design. This conclusion is further supported by the visuals presented in Figures 17–21 and 23–29. Therefore, it can be concluded that wood and metal waste materials possess considerable potential for incorporation in interior design, particularly in terms of enhancing aesthetics, functionality, and sustainable practices.

Figure 17 showcases a storage cupboard designed for students of the Art Education Department, constructed using wood waste. The images demonstrate the efficient utilization of space when the cupboard is folded, highlighting its compactness. Additionally, the figures depict the inclusion of a work table and chair that can be conveniently stored within the cupboard.

The utilization of technical materials derived from wood and metal waste in the interior design of spaces has been observed to a significant extent. This signifies that these waste materials can be repurposed and recycled in a manner that promotes both environmental and aesthetic sustainability. These results and the executed artistic works indicate that wood and metal waste, as representative raw materials, contribute unique aesthetic values, as shown in Figures 17 and 25. They have been incorporated into the design process in line with design principles and elements such as rhythm, repetition, balance, contrast, color usage, and the integration of diverse materials like fabrics and other artistic elements, as shown in Figures 18 and 19, all tailored to suit the nature of the interior spaces.

Moreover, the furniture elements designed by the students demonstrate a comprehensive understanding and awareness of their environment, its preservation, and the effective utilization of its various components to create functional furniture elements. These achievements align with the pursuit of sustainable development goals across various fields, particularly in artistic specializations like interior design. This generates creative ideas, fosters originality, and establishes connections with heritage, while also showcasing the flexibility in utilizing and synthesizing multiple materials derived from wood and metal waste, thereby unleashing imaginative potential to harmonize this heritage with the educational aspect, as shown in Figures 20, 21 and 26–28.

The involvement of students in the implementation of these works revealed several positive aspects related to enhancing cultural values, achieving sustainability, and protecting the environment from visual distortions. Furthermore, these works elevated the level of performance and artistic, sensory, and national beauty, which had a noticeable impact on the responses and participation in evaluating furniture elements.

These findings are consistent with previous studies, such as the research conducted by Bahloul and Sarah [11], which highlighted the achievement of social balance, as well as the study by Sara et al. [8], emphasizing the significant role of interior designers in environmental preservation through sustainable designs and their ability to showcase the identity and character of different environments, including the use of recycled materials such as wood and metal waste. These sustainable materials are highly desirable for utilization within Al-Ahsa's natural environment and can be employed in various settings.

In relation to the third question, the researchers investigated whether there were statistically significant differences, at a significance level of $\alpha \geq 0.05$, between the aesthetic dimension and the functional dimension of furniture elements created by art education students in the interior design course.

To examine this hypothesis, the *t*-test was employed to determine the statistical significance of sustainability standards associated with the aesthetic and functional dimensions of the furniture elements implemented by the students. Specialists analyzed the analysis card axes of the artistic product, focusing on the aesthetic and functional dimensions. They evaluated the effectiveness of the furniture elements and assessed differences in sustainability within the interior spaces. The results of this analysis are presented in Table 2, which provides further insights into these findings.

The results of the *t*-test, presented in Table 2 and Figure 22, revealed significant differences between the mean scores of the sustainability criteria for the implemented furniture elements and the evaluations provided by the specialists. The data in the table indicate that the probability value of the *t*-test in all areas was below the predetermined error level of $\alpha$ ($0.05 \leq \alpha$). This suggests that the technical products' sustainability standards were highly achieved in their design utilization.

**Table 2.** Significance of differences in average scores of specialists on technical product analysis cards for aesthetic and functional sustainability dimensions of wood and metal materials in interior design.

| Dimensions of Sustainability | Mean | Standard Deviation | N | Degree of Freedom | t-Value | Sig. |
|---|---|---|---|---|---|---|
| Aesthetic dimension | 4.026 | 0.7243 | | | 5.878 | |
| Functional dimension | 3.943 | 0.6154 | 15 | 14 | 6.767 | 0.036 |
| Overall degree of verification | 3.984 | 0.6670 | | | | |

Specifically, the aesthetic dimension exhibited a higher arithmetic mean (4.026) compared to the functional dimension (3.943). This indicates that the specialists' responses in analyzing and evaluating the students' furniture elements were highly favorable in terms of aesthetics and functionality. This success can be attributed to the utilization and adaptation of wood and metal waste materials in creating functional furniture elements. The analysis, depicted in Figures 17–21 and 23–29, highlights the positive aspects of design, implementation, diversity, innovation, and modernity resulting from the use of wood and metal waste.

The high degree of achievement of sustainability standards in interior design within the analyzed interior spaces can be attributed to the students' strong abilities in aesthetics and functionality, as well as their connection to their homeland and environment. The utilization of waste materials enhances the feeling of conservation and protects the environment from visual distortions and pollution. By encouraging the utilization of various environmental raw materials, especially wood and metal waste, in different furniture components, new and innovative approaches are employed, which enhances the economic and professional aspects among various segments of society.

The utilization of wood and metal waste in interior design also plays a crucial role in shaping behaviors and improving individuals' attitudes. It reduces the likelihood of future visual distortions in the environment by developing artistic taste and community awareness regarding the utilization and reuse of environmental waste. This responsibility towards preserving the environment arises from the realization that spontaneous beauty originates from the environment itself, while cognitive beauty stems from human consciousness. These findings align with previous studies conducted by Naseer [12] and Al-Kandari [13], which emphasized the potential for improving students' innovative skills in utilizing waste materials and modifying the behavior and awareness of others towards the environment.

The researchers of this study believe that employing recycling techniques for wood and metal waste is a crucial method for environmental preservation. It helps in energy conservation, mitigates visual distortions, contributes to economic growth, and offers rational solutions for utilizing waste to fill interior spaces that align with design components in educational settings. This creates an environment based on aesthetic, artistic, and functional foundations, ultimately enhancing students' taste levels. Recycling waste also contributes to achieving social–environmental balance, creating employment opportunities, and developing effective strategies for waste treatment and recycling, thereby positively impacting sustainable development dimensions.

## 4. Conclusions

In conclusion, the findings from the tables above provide evidence supporting the utilization of wood and metal waste as artistic materials in interior design, along with their potential for recycling. The criteria incorporated in the scale demonstrated that the students' engagement with recycled furniture elements was enhanced through their behavior, innovations, and skills. Furthermore, the artistic products showcased the successful attainment of aesthetic and functional dimensions in interior design.

The results of this study suggest that as students or individuals gain exposure to educational skills and knowledge, as well as develop an awareness of their environment and their connection to it, their behavior and aesthetic perception undergo positive transformations. This, in turn, enhances their artistic creativity and their ability to effectively

utilize environmental waste for recycling purposes. Consequently, this has a significant impact on the artistic product produced. Figure 29 presents a visual representation of several completed works created using wood and metal waste materials.

The findings of this study indicate that there is a social interaction among students, with a focus on knowledge, skills, different thinking approaches, and strategies related to the utilization and employment of wood and metal waste in interior design. This has led to the development of aesthetic and functional sensibilities by challenging traditional methods in education and the evaluation and appreciation of artistic work.

The current study has contributed to increasing designers' awareness of various techniques for recycling wood and metal waste, providing them with multiple design solutions to enhance the aspects of design and interior space.

**5. Future Directions**

This study, which explores the potential of utilizing recycled wood and metal waste in interior design applications for environmental sustainability, can serve as a foundation for future research. The following potential directions for future studies are suggested:

Long-term follow-up: Conducting a longitudinal study to assess the long-term effects of art-based interventions on interior design applications and related outcomes. This would help determine whether the positive effects observed during this study persist over time.

Comparative studies: Comparing the effectiveness of art-based interventions with other intervention methods or materials to evaluate their relative efficacy. This could involve comparing the outcomes of art-based interventions with traditional counseling, design and environmental interventions, or other creative approaches.

Diverse participant groups: Expanding this study to include diverse participant groups such as individuals interested in interior design, architecture, interior spaces, and interior designers from different cultural backgrounds or varying socioeconomic statuses. This would provide insights into the effectiveness of art-based interventions across different contexts.

Mechanisms of change: Investigating the underlying mechanisms through which art-based interventions influence the overall awareness of individuals studying in educational settings. This could involve exploring how engagement in artistic activities affects educational motivation, self-expression, or social–environmental skills.

Implementation and scalability: Exploring the feasibility and practicality of implementing art-based interventions in real-world educational settings, such as classrooms, breakout areas, or community centers. Evaluating factors such as training requirements, resource availability, and potential implementation barriers can inform the scalability and sustainability of such interventions.

Measurement and evaluation tools: Developing and validating evaluation tools to measure changes in wood and metal waste utilization in design and related outcomes resulting from art-based interventions. This would contribute to the field by providing reliable and valid measures for evaluating the effectiveness of similar interventions in different contexts.

**6. Recommendations**

- Reconsider the utilization of wood and metal waste in the Al-Ahsa region based on a specific national strategy that aligns with the educational context and aims to minimize visual distortion. This should be carried out through a comprehensive visual study that assesses the potential uses of waste materials in a way that enhances visual aesthetics.
- Focus on developing behaviors as a key aspect in addressing visual distortion, thereby reducing the likelihood of its occurrence in the future. This can be achieved by elevating the level of artistic appreciation and actively involving various educational institutions in fostering artistic awareness among the educational community members.

- Identify the different types of waste components in the Al-Ahsa region, assess their quantity and production rates, and develop effective programs for their disposal and future planning to promote environmental sustainability through their utilization.
- Promote workshops, seminars, and scientific conferences dedicated to the development and recycling of wood and metal waste, with a specific focus on their connection to decoration and interior design. This will contribute to advancing environmental sustainability efforts.
- Recognize that interior design should not be detached from the advancements in modern science and recycling techniques. It should maintain an interactive and integrated approach with research in the field of recycling and the use of waste materials in various settings.
- Conduct further specialized studies to explore the production of raw materials derived from wood and metal waste, which can be readily utilized in interior design beyond educational settings and in various locations.

**Author Contributions:** All authors have sufficiently contributed to this study and agreed with the results and conclusions. All authors have read and agreed to the published version of the manuscript.

**Funding:** This work was financially supported by the Deanship of Scientific Research, King Faisal University, Saudi Arabia (grant number GrantA269).

**Institutional Review Board Statement:** This study was conducted in accordance with the Declaration of Helsinki and approved by the Ethics Committee of King Faisal University (protocol code: ERS_2022_6205 and date of approval: 28 November 2022).

**Informed Consent Statement:** Informed consent was obtained from all individual participants included in this study.

**Data Availability Statement:** The data supporting this study's findings and conclusions are available upon request from the corresponding author.

**Conflicts of Interest:** The authors declare no conflicts of interest.

## Appendix A

**Table A1.** Analysis card for aesthetic and functional dimensions of sustainability for utilization wood and metal waste in interior design.

| No. | Dimension and its Items | Excellent | Very Good | Good | Acceptable | Low |
|-----|-------------------------|-----------|-----------|------|------------|-----|
| | Aesthetic dimension | | | | | |
| 1 | Characteristics of the utilized material in the artwork | | | | | |
| 2 | Diverse employment of elements and principles of artistic work | | | | | |
| 3 | Incorporation of multiple materials in the artwork for enhanced diversity | | | | | |
| | Functional dimension | | | | | |
| 1 | Proficiency in executing intricate details of technical work | | | | | |
| 2 | Alignment of form with function | | | | | |
| 3 | Precision and accuracy in the execution of artwork details | | | | | |

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
