# Peer review of "Exploring the Aesthetic and Functional Aspects of Recycled Furniture in Promoting Sustainable Development: An Applied Approach for Interior Design Students"

_sustainability, doi:10.3390/su16104003_

Round 1
Reviewer 1 Report
Comments and Suggestions for Authors
This study explores the aesthetic and functional dimensions of recycled furniture in promoting sustainable development. The findings suggest that these pieces have reached significant levels of sustainability in both aesthetic and functional dimensions. Since the works have already incorporated recycled materials, what is the additional significance of probing into the works' sustainability?
Specifically, concerning the six criteria on the analysis card: Characteristics of the utilized material in the artwork, Diverse employment of elements and principles of artistic work, Incorporation of multiple materials in the artwork for enhanced diversity, Proficiency in executing intricate details of technical work, Alignment of form with function, Precision and accuracy in the execution of artwork details—these do not have a direct relationship with sustainability. Though an indirect relationship may exist, it could lead to a diversion from the main focus of the research inquiry. This study needs to adequately address the number and background of the respondents to the analysis card. The statistical treatment in section 2.5 is superfluous.
In summary, the statistical analysis that led to the average scores for the aesthetic and functional dimensions and their significance does not have a direct link to sustainability. The analysis card may be fitting or beneficial for evaluating interior design works. Using recycled materials for artistic creation is a choice, yet this choice offers limited contributions to the advancement of sustainability research.
Author Response
Response |
Comments |
No. |
The importance of the study has been modified |
This study explores the aesthetic and functional dimensions of recycled furniture in promoting sustainable development. The findings suggest that these pieces have reached significant levels of sustainability in both aesthetic and functional dimensions. Since the works have already incorporated recycled materials, what is the additional significance of probing into the works' sustainability? |
1 |
The researchers employed six criteria to assess the sustainability of materials used in the study, such as wood and metals including recycled welds, to evaluate their suitability for designed furniture elements. These criteria were applied to measure sustainability and assess the response of recycled materials based on the researchers' perspective. |
Specifically, concerning the six criteria on the analysis card: Characteristics of the utilized material in the artwork, Diverse employment of elements and principles of artistic work, Incorporation of multiple materials in the artwork for enhanced diversity, Proficiency in executing intricate details of technical work, Alignment of form with function, Precision and accuracy in the execution of artwork details—these do not have a direct relationship with sustainability. Though an indirect relationship may exist, it could lead to a diversion from the main focus of the research inquiry. This study needs to adequately address the number and background of the respondents to the analysis card. The statistical treatment in section 2.5 is superfluous. |
2 |
Furthermore, this aspect builds upon the preceding point, as the researchers assert that beauty and functionality are the fundamental dimensions of the study and form the basis of the analysis framework. While there are undoubtedly other criteria that contribute to sustainability and offer further insights into sustainability research, this study specifically focused on these two dimensions. The researchers express gratitude for the comments provided by the referee, as they have positively influenced the quality and enrichment of the research. |
In summary, the statistical analysis that led to the average scores for the aesthetic and functional dimensions and their significance does not have a direct link to sustainability. The analysis card may be fitting or beneficial for evaluating interior design works. Using recycled materials for artistic creation is a choice, yet this choice offers limited contributions to the advancement of sustainability research. |
3 |
Reviewer 2 Report
Comments and Suggestions for Authors
The article concludes the feasibility of recycled wood and metal in terms of aesthetic and functional dimensions related to sustainability through the marking and assessment by professionals of furniture pieces designed by students using recycled wood and metal. The paper raises meaningful research questions that are fully explored, and the considerations and reviews are thorough before the experiment. The structure of the paper is logically clear and the sections are appropriately titled. However, the details are not clear enough. There is no basis for the selection of the experimental samples. and the number of people who scored the experimental samples was not convincing. To ensure the quality of the article, it is recommended to make modifications and supplements to some of the content mentioned below.
1. In 1.9 Theoretical framework, The content in Table 5-9 are furniture pieces designed and made using recycled metal and wood, which are more suitable to be used as display examples in the Sixth: Technical uses of wood and metal waste. It is more suitable to be supported in the seventh heading with literature that examines technical experiments.
2. In 2.2. Study Population and Sample, “This sample was deemed representative of the broader student body within the Department of Art Education.”.What is the foundation for selecting a representative sample of 11 designs?
3. In 2.3.3. Correcting the sustainability scale. Why did you choose three professionals to score the artwork? Too few people scoring the artworks may result in less accurate data results.
4. Figure 16. It would be more intuitive to change the numbers on the blue statistic portion of the bar chart to black or dark color, or to label the numbers outside the rectangles. The white numbers are now in a position that makes them unclear.
5. In 3. Results and Discussion, replace "Figures 21-17" with "Figures 17-21" for better readability.
6. In 3. Results and Discussion, "Figures 21-17 and 23-30" show students' designs using recycled materials, with different shooting angles, different proportions of pictures, and backgrounds that do not make use of recycled materials. It is recommended to simplify the backgrounds, unify the shooting angles, unify the proportions of pictures, and arrange the pictures neatly.
Comments on the Quality of English Language
Extensive editing of English language required.
Author Response

(The authors gave the same response as above.)

Reviewer 3 Report
Comments and Suggestions for Authors
After reviewing the manuscript 'Exploring the Aesthetic and Functional Aspects of Recycled Furniture in Promoting Sustainable Development: An Applied Approach for Interior Design Students', I conclude that the research presented in the paper on the use of environmental art as a strategy for preserving and fostering sustainable development is relevant. It is also relevant to investigate the economic benefits of investing in waste recycling and environmental sustainability, highlighting the potential for positive returns by engaging students in creative design practices and using green raw materials, which was done during the study. The paper contains all the necessary parts; the results obtained are valuable and allow make evidence-based conclusions. The authors qualitatively discussed all the issues determined by the problems of environmental sustainability, giving useful insights for the development future sustainability strategies. While emphasizing the positive impression made by the article, it must be said that several flaws were found. I think that these points should be communicate better.
In the Chapter 1.8 Literature Review information about the work done by other authors is presented in too much detail. The authors should reviewed the current state of the research field instead of retelling other authors‘articles.
Chapter 1.9 is too long; it looks like it could be a separate article on its own, It is not clear how the information are related with research carried out?
In Chapter 2. Methods I did not find how the agreement between expert was evaluated? Was the coefficient of concordance calculated? It is stated that “These models were then subjected to analysis by experts in the field“ (line 788). How this analysis was carried out? Give necessary details to perform such analysis. What qualification requirements are imposed on experts, just so that they work in the fields of art and interior design?
In 3. Results and Discussion there is no need to duplicate the information, table 1 or Fig.16 (which needs to be arranged qualitatively) is enough. How can be Contribution level understand (Table 1)?
It is not possible to agree with the statement that ‚ “Visual representations of the findings can be observed in Figures (17-30) and diagram Figure 16“ (line 791). The figures should be much more commented, not leaving the reader to interpret what is depicted in them, whether they adequately illustrate the idea being explored. An analogous situation is with „This conclusion is further supported by the visuals presented in Figures 21-17 and 23-30.(Line 821)
I suggest rethinking the keywords so that they be more specific to the article
Author Response

(The authors gave the same response as above.)
